# Lifestyle Risk Factors for Overweight/Obesity in Spanish Children

**DOI:** 10.3390/children9121947

**Published:** 2022-12-12

**Authors:** María L. Miguel-Berges, Pilar De Miguel-Etayo, Alicia Larruy-García, Andrea Jimeno-Martinez, Carmen Pellicer, Luis Moreno Aznar

**Affiliations:** 1Growth, Exercise, Nutrition and Development (GENUD) Research Group, University of Zaragoza, C/Domingo Miral s/n, 50009 Zaragoza, Spain; 2Instituto Agroalimentario de Aragón (IA2), C/Miguel Servet 177, 50013 Zaragoza, Spain; 3Instituto de Investigación Sanitaria de Aragón (IIS Aragón), Avda. San Juan Bosco 13, 50009 Zaragoza, Spain; 4Centro de Investigación Biomédica en Red-Fisiopatología de la Obesidad y Nutrición (CIBEROBN), (CB15/00043), Institute of Health Carlos III (ISCIII), Av. Monforte de Lemos, 3-5, Pabellón 11, Planta 0, 28029 Madrid, Spain; 5Fundación Trilema, Carrer d’Isaac Peral i Caballero, 9, 46980 Paterna, Spain

**Keywords:** childhood obesity, risk factors, healthy lifestyles

## Abstract

Childhood obesity is one of the main public health concerns in Europe. The aim was to identify possible risk factors associated with overweight/obesity in Spanish preschool and school-age children. The sample (1075 (50.7% girls) children aged 3 to 12) is part of the project ‘Alimentando el Cambio’ whose objective is to promote healthy lifestyles in schools. Child height and weight were measured, and parents filled out questionnaires related to the children’s lifestyle. There was a positive and significant association between sweetened beverage consumption and body mass index (BMI) z-score in both sexes and age groups. There was a negative and significant association between BMI z-score and dairy products in girls of both age groups. There was also a protective effect of regular nut consumption on overweight/obesity in girls 6–12 y. Night-time sleep during weekdays showed a negative association with BMI z-score for older boys and girls. A positive and significant association was found between total screen time and BMI z-score during weekdays. Regarding emotional well-being and self-esteem, having girls 6–12 y laughing and feeling happy and good about themselves in the last week was a protective factor against overweight/obesity. Childhood obesity prevention efforts may benefit from targeting these key risk factors.

## 1. Introduction

Childhood obesity is one of the main public health concerns in Europe; according to the last report from the ALADINO study, in 2019, the prevalence of overweight and obesity in schoolchildren in Spain was 23.3% and 17.3%, respectively [1]. Children and adolescents with overweight/obesity have a high probability of maintaining excess body fat mass in adulthood [2]. Moreover, the presence of obesity is associated with numerous health complications that have been found early in life [3], such as type 2 diabetes and cardiovascular diseases [4,5].

Obesity is a multifactorial condition [6]. There are several factors that contribute to overweight and obesity development [7]. Genetics is one of the main predictors of overweight in children and adolescents [8]. On the other hand, dietary factors such as low fruit or vegetable intake [9,10], high sugar-sweetened beverages consumption [11] and high fat intake [12] have been associated with high obesity prevalence. However, foods are consumed in a whole dietary pattern, which seems to be relevant in relation to health outcomes [13]. In addition to these dietary patterns, there are some methods, such as several diet quality indices, to assess adherence to dietary recommendations [14,15,16]. One of them is the Mediterranean diet adherence (MedDiet), which has been related to central obesity, hypertriglyceridemia and insulin resistance in 9–13-year-old school children [17]. An interaction between genetic predisposition and Mediterranean diet adherence has also been observed [8].

On the other side of the energy balance, low levels of physical activity and sedentary and sleep behaviours play an important role in the development of overweight/obesity. In longitudinal studies, daily physical activity—at least 60 min per day of moderate/vigorous physical activity (MVPA) [18]—sedentary behaviours [19] and sleep duration [20] have been shown to affect energy balance from early childhood [21,22].

Weight-related impairment in the emotional well-being of the young population has increased significantly in obesity research [23,24]. The association between obesity and emotional well-being has shown inconsistent findings in children, adolescents and adults [25,26,27]. However, obesity has been found to be associated with low self-esteem and depressive mood in some studies [23,28,29]; in other studies, no association has been observed [24,30,31]. Recently, the effects of weight-related body dissatisfaction in boys and girls during adolescence have been studied [32]. This situation is associated with negative body image, which may be a key approach in obesity prevention and treatment programmes.

Lifestyle is a summary of the behaviour patterns that we develop during our socialisation processes, which occur during preschool and school age. These behaviours are learned in relationships with parents, peers, friends and siblings or through the influence of school or the media. These patterns include children’s physical activity, screen and sleep time, beverages and food consumption. On the other hand, our lifestyle is also influenced by our emotional well-being and self-esteem as the main psychological and emotional factors to ensure a balance that supports daily decisions.

The topic included in the manuscript has been widely studied, but there is no report on all the potential lifestyle determinants together, including emotional well-being and self-esteem, in a Spanish sample whose methodology is innovative. Therefore, the objective of this study was to identify the most relevant risk factors associated with overweight/obesity in Spanish preschool and school-age children.

## 2. Materials and Methods

### 2.1. Participants

The current study used baseline data from the programme ‘Alimentando el cambio’, a longitudinal study of preschool and school-age children recruited from 6 schools in Spain. The programme is named Fluye (https://www.proyectofluye.com/ (accessed on 31 October 2022)), and the objective was to promote healthy lifestyles in selected schools belonging to the Trilema Foundation. The school participation rate was 69.2%, and a total of 1075 (50.7% girls) children aged 3 to 12 years were included (participation rate = 64.3%) (Figure 1). For this analysis, a cross-sectional design was performed. Only baseline data collected from January 2020 until March 2020 were included. Information regarding preschool and school-age children’s physical activity, screen and sleep time, beverage consumption, Mediterranean diet adherence, emotional well-being and self-esteem and sociodemographic and socioeconomic characteristics were obtained via parental self-reported questionnaires (father, mother, grandparents, among others). Parents or legal guardians gave written informed consent for the examination of their children. Ethical approval was obtained from the Aragon Committee of Ethics in Clinical Research (CEICA).

### 2.2. Socioeconomic Variables

The questionnaire included a question about years of education and professional qualifications of both parents. Maternal and paternal education levels (the years of education) were grouped into four categories: no studies, basic studies, professional formation and university formation. Maternal education was identified as one of the best proxy indicators of familiar socioeconomic status [33] and was used for the analysis.

### 2.3. Anthropometric Measures

Anthropometric measures were performed by trained researchers according to ISAK Protocol [34]. Body weight was measured in underwear and without shoes using an electronic scale (Type SECA 877 (2–200kg, SECA, Hamburgo, Alemania) to the nearest 0.1 kg, and body height was measured with a telescopic height instrument (Type SECA 213 (Hamburgo, Alemania) to the nearest 0.1 cm. Body mass index (BMI) was calculated as weight in kg divided by height in metres squared. BMI z-score was calculated according to Cole et al. [35]. Moreover, waist (WC) circumference was measured to the nearest millimetre with an inelastic tape (Cescorf, Brasil), with the subject standing upright.

### 2.4. Beverage Consumption

Beverage consumption was assessed using an 11-item semiquantitative frequency questionnaire extracted from the validated food frequency questionnaire used in the ToyBox study [36] (Appendix A). In the current study, eight beverage groups were selected and analysed: (1) water, (2) soft drinks (sugar-sweetened and light beverages), (3) fresh fruit juices and packed juices, (4) sugar-sweetened beverages (soft drinks and juices), (5) smoothies, (6) milk, (7) yoghurt and (8) dairy products (milk and yoghurt).

### 2.5. Mediterranean Diet Adherence

Adherence to the Mediterranean diet was assessed using a validated questionnaire [37] (Appendix A). The questionnaire includes 12 questions on food consumption frequency and 2 questions on food intake habits considered characteristic of the Spanish Mediterranean diet. Each question was scored 0 or 1. If the condition was not met, 0 points were recorded for the category. The final score ranged from 0 to 14. Then, the score was recategorised: scores < 3 were considered a diet very far from a Mediterranean diet model, scores 3–7 were considered an acceptable diet, although they required improvement, and scores > 8 were considered an adequate diet.

### 2.6. Physical Activity

PA was assessed by a questionnaire through sports participation (number of hours per week that children participated in one or two sports) (Appendix A). The assessment of PA through ‘sports participation’ was identified in previous European studies as showing the highest correlation with moderate-to-vigorous PA as measured with accelerometers [38].

### 2.7. Screen Time

Total screen time (TST) (i.e., television and computer time) was assessed, both for week and weekend days, by two questions: (1) minutes spent watching television (including video and DVD) and (2) minutes spent on computer activities per day. Responses included were ‘never’, ‘<30 min/d’, ‘30 min to 1 h/d’, ‘1–2 h/d’, ‘3–4 h/d’, ‘5–6 h/d’, ‘7–8 h/d’, ‘8 h/d’ and ‘more than 8 h/d’. Then, the categorical variables were transformed into minutes per day for week and weekend days separately. In addition, these answers were further aggregated into two categories, including ≤1 h/d and >1 h/d in the case of preschool children and ≤2 h/d and >2 h/d in the case of school-age children. These categories are based on the World Health Organization [39] and Canadian sedentary behaviour recommendations [40].

### 2.8. Sleep Duration

Sleep time was reported by parents through the number of hours and minutes the child slept per night on average. They were reported separately for weekdays and weekend days.

### 2.9. Emotional Well-Being and Self-Esteem

Psychosocial well-being was assessed with questions on emotional well-being, self-esteem and family life based on three subscales of the ‘KINDLR questionnaire’, a validated instrument for measuring health-related quality of life in children and adolescents [41] (Appendix A). The questionnaires included 12 questions that were filled out by the parents. Responses included were: ‘not at all’, ‘almost never’, ‘sometimes’, ‘often’ and ‘always’.

### 2.10. Statistical Analysis

Predictive Analytics Software version 20 (IBM SPSS Statistics for Windows) was used to analyse the data. Statistical analysis was stratified by sex and age (3–6 y vs. 6–12 y), and all analyses were adjusted for maternal education. Initially, the variables were analysed using a *t*-test for continuous variables and χ^2^ test for categorical variables. Analysis of covariance (ANCOVA) was used to investigate differences between each total screen time category for the BMI z-score. Additionally, linear and logistic regression models were performed to identify the odds of having overweight/obesity between the different variables of children’s lifestyles with ‘yes’ and ‘always’ as the reference groups. All statistical tests and corresponding *p*-values lower than 0.05 were considered statistically significant.

## 3. Results

Table 1 shows descriptive information about the mean and standard deviation of age, BMI z-score, waist and hip circumference and waist/height index, as well as the number and percentage of the sample according to the BMI categories and maternal education. There were significant differences between sexes regarding BMI z-score and waist circumference.

Analysis derived from linear regression is shown in Table 2. There was a positive and significant association between soft drinks consumption and BMI z-score in boys (6–12 y). When soft drinks and juices were merged, it was observed a positive and significant association between sugar-sweetened beverages and BMI z-score in boys and girls of both age groups. On the contrary, there was a negative and significant association between milk consumption in girls 3–6 y and dairy products (milk and yoghurts) in girls of both age groups.

In Table 3, individual components of the Mediterranean diet adherence questionnaire were analysed. There was only an increased odds of having overweight/obesity when girls 6–12 y did not consume nuts regularly.

Table 4 shows the analysis of covariance for BMI z-score according to total screen time categories. Negative and significant associations were found for night-time sleep during weekdays and BMI z-score for both boys and girls aged 6–12 y. In boys from both age groups, there was a negative and significant association for night-time sleep duration during weekend days. A positive and significant association was found for total screen time and BMI z-score during weekdays for boys 3–6 y and girls 6–12 y. Only for boys 3–6 y was there a negative and significant association with total screen time during the weekend.

Regarding emotional well-being and self-esteem (Table 5), there was an increased odds of having overweight/obesity when girls 6–12 y were laughing and feeling happy and feeling good about themselves in the last week. On the opposite, there was a protective effect of overweight/obesity when parents of 6–12 y boys reported that their child was feeling overprotected by their parents in the last week.

## 4. Discussion

A large set of childhood risk factors for obesity in a key developmental period has been considered in this study. We found that sugar-sweetened beverages consumption, total screen time and feeling overprotected by their parents were significantly associated with childhood overweight/obesity. Conversely, the consumption of nuts and dairy products, sleep duration and feeling happy and also feeling good about themselves were factors protecting against obesity.

### 4.1. Beverage Consumption and Obesity

The positive association between sugar-sweetened beverages (SSB) and overweight/obesity found in our study is consistent with the literature. Two meta-analyses from 2013 found a positive association between SSB consumption and weight gain in children and adults [42,43]. In the same way, a systematic review of prospective cohort studies and intervention trials suggested that SSB consumption had an effect on obesity indices in children and adults [44,45]. Some authors have suggested that compensation at subsequent meals for energy consumed in the form of a liquid is less complete than for energy consumed in solid form [46]. This has also been observed in an experiment carried out by Bellisle et al. [47], in which the consumption of sweetened drinks as long as 1 h prior to eating suppressed food intake, reducing diet variety. A study simulated the effect of replacing one serving of SSB (~0.25 L) with one serving of water and estimated a decrease in energy intake from beverages from 17% to 11%, predicting a reduction in the prevalence of obesity [48].

### 4.2. Mediterranean Diet and Obesity

Our results revealed a protective effect against obesity related to the consumption of nuts. Nuts are considered an energy-dense food [49], but nonetheless, have not been associated with weight gain in epidemiological studies; most experimental nut feeding studies reported no association with weight gain [50,51]. This fact might be explained because of the elevated energy expenditure of unsaturated fat [49,52]. Furthermore, the high content of dietary fibre and proteins in nuts explains their high satiety rating [49]. The Mediterranean diet (MD), described in the 1960s, is considered a dietary pattern characterised by a high consumption of foods of vegetable origin, such as fruits, vegetables, legumes, nuts, cereals and olive oil, and, on the other hand, a low consumption of meat and processed meat [53,54]. In Spanish adults, an inverse association between the Mediterranean diet and BMI was reported [55]. In Spanish children (PREFIT study) aged 3–5 years, low adherence to the MD was associated with high waist circumference and BMI [56]. In the same way, a recent study revealed that ‘poor’ adherence to the MD was associated with an increased likelihood of obesity in a representative sample of Greek schoolchildren [17]. A cross-sectional study among 1140 children (9–13 y) carried out in Cyprus concluded that adherence to the MD was inversely associated with obesity [57]. A Greek study (the Greco study) also reported that abandonment of the Mediterranean diet among children was related to obesity [58].

### 4.3. Total Screen Time (TST) and Obesity

In our study, TST was identified as a risk factor for overweight/obesity in preschool and school-age children. A recent systematic review with meta-analysis found that TST, television time and computer time were associated with overweight/obesity among children; the main conclusion was that total screen time of ≥2 h/day could be one of the most critical risk factors for overweight/obesity in children and adolescents [59]. In a multicentre study of children (9–12 y) from 12 countries all over the world, a high prevalence of obesity was observed in those children spending more time in front of screens [60]. In numerous studies, high levels of screen time have been associated with a greater frequency of consumption of energy-dense, low-micronutrient products, a low frequency of consumption of fruits and vegetables and with overweight and obesity in children [61,62,63,64] and adolescents [65].

### 4.4. Sleep Duration and Obesity

An inverse association between night-time sleep duration and overweight/obesity risk in preschool and school-age children was revealed. This result is consistent with the current literature [20,66,67,68,69]. A meta-analysis concluded that current studies from around the world show that short sleep duration is consistently associated with the development of overweight/obesity in children and young adults [70]. In Spanish adolescents, it was observed that almost 45% of adolescents in Spain are not sleeping the recommended amount of time; in addition, a short duration of sleep was associated with obesity [71].

### 4.5. Emotional Well-Being and Self-Esteem

In our results, body dissatisfaction was associated with obesity in school-age girls. Davison et al. [72] observed that 9-year-old girls tend to report lower body esteem and lower perceived cognitive ability, and this has been associated with obesity. Paxton et al. [73], in a review of the topic, concluded that negative body image attitudes were related to the onset of disordered eating, poor self-esteem, general mental health problems and obesity. This could be related to some findings in our study, such as school-age girls who did not feel happy in the last week had an increased likelihood of having overweight/obesity. A multicentre study performed in Germany, UK and Australia showed that in all three countries, the low subjective well-being of individuals had a positive association with obesity [74]. A study performed on school-age children showed a relationship between low self-esteem levels and overweight and obesity. In relation to the question ‘Am I a happy person?’ a negative response to the question was associated with excess weight [75].

### 4.6. Limitations and Strengths

This study has some limitations as well. Information on diet and screen time was collected via parental self-reported questionnaires, which are prone to over- or underreporting. This study has several strengths, including the standardised procedures used for collecting anthropometric measurements and detailed beverage information from semiquantitative food frequency questionnaires, a methodology that has been widely used in European studies [36]. On the other hand, Mediterranean diet adherence was measured with an easy and validated questionnaire in preschool and school children. Another strength includes the inclusion of children recruited from different districts and six different schools in Spain.

## 5. Conclusions

In order to design obesity intervention programmes, it is necessary to elaborate on key messages related to lifestyle behaviours and subjective well-being. This study identified specific influences in early life that might be suitable targets for childhood obesity prevention efforts.

Sugar-sweetened beverage consumption in both sex and all age groups, total screen time in preschool boys and girls (6–12 y) and feeling overprotected by their parents in only girls (6–12 y) was associated with childhood obesity. In contrast, the consumption of nuts and dairy products in girls, sleep duration in boys and girls (6–12 y) and feeling happy and good about themselves in only girls (6–12 y) were associated with a protective effect against obesity. Therefore, childhood obesity prevention efforts may benefit from targeting these key risk factors.

## Figures and Tables

**Figure 1 children-09-01947-f001:**
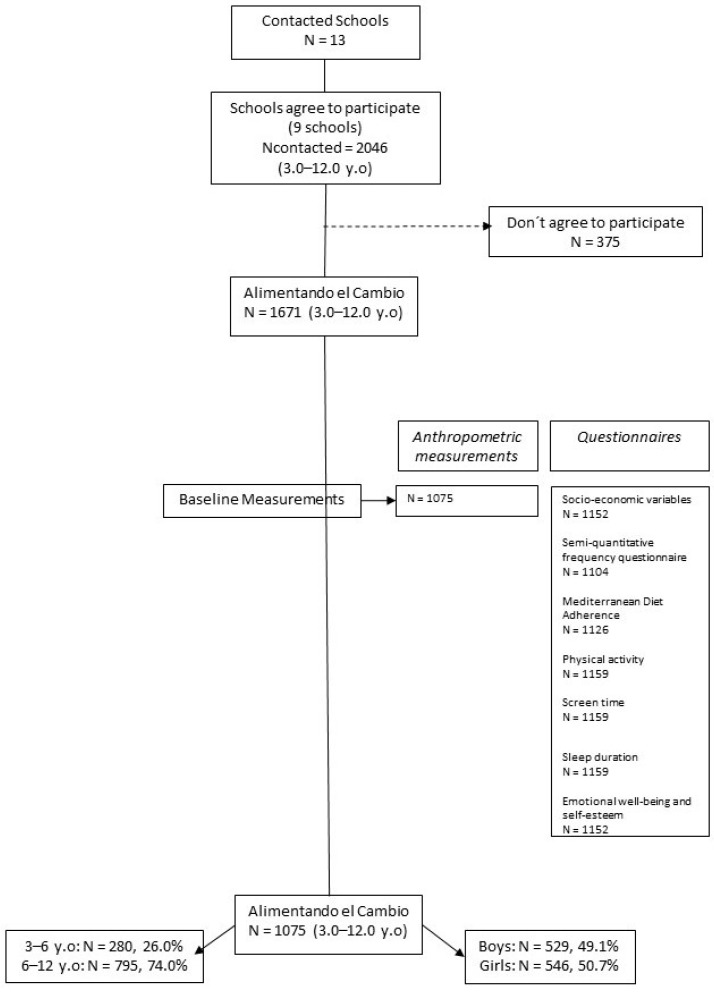
Flow chart of the participants involved in this study from ‘Alimentando el cambio’.

**Table 1 children-09-01947-t001:** Descriptive characteristics of the sample participating in the study (*n* = 1075).

	Boys (*n* = 529; 49.1%)	Girls (*n* = 546; 50.7%)	
	Mean (SD)	Mean (SD)	*p*
Age (years)	7.96 (2.76)	8.08 (2.69)	0.305
BMI (z-score) *	0.71 (1.19)	0.50 (1.15)	**0.006**
Waist circumference (cm)	60.84 (9.09)	59.53 (8.86)	**0.018**
	*n* (%)	*n* (%)	
BMI (categories) **			
Thinness	17 (3.0)	18 (3.0)	0.825
Normal weight	344 (68.9)	347 (69.1)	
Overweight	95 (19.0)	97 (19.3)	
Obesity	43 (8.6)	40 (8.0)	
Maternal education			
Compulsory studies	101 (17.01)	124 (20.7)	0.163
Professional formation	306 (51.17)	186 (31.01)	
University studies	179 (29.9)	288 (48.16)	

Sex differences using Pearson’s chi-square test for categorised variables and *t*-test for continuous variables. Significant differences (*p* < 0.05) are shown in bold font. * BMI (body mass index) z-score was calculated according to the cut points of Cole et al. [24]. ** BMI (body mass index) was categorised according to the cut points of Cole et al. [24].

**Table 2 children-09-01947-t002:** Association between beverage consumption and BMI z-score (*n* = 1075).

	3–6 y				6–12 y			
Portions/Day	Boys (*n* = 139; 13%)		Girls (*n* = 141; 13.2%)		Boys (*n* = 390; 36.2%)		Girls (*n* = 405; 37.6%)	
	B	*p*	B	*p*	B	*p*	B	*p*
Water	0.06	0.567	0.18	**0.042**	−0.03	0.637	−0.036	0.592
Soft drinks	−0.03	0.921	−0.13	0.906	0.44	0.013	0.320	0.121
Juices	0.03	0.838	−0.12	0.400	0.09	0.275	−0.500	0.624
Sugar-sweetened beverages *	0.317	**0.035**	0.151	**0.040**	0.179	**0.026**	0.089	**0.043**
Smoothies	0.07	0.417	−0.02	0.876	−0.02	0.841	0.106	0.138
Milk	−0.15	0.148	−0.09	0.296	−0.05	0.524	−0.19	**0.032**
Yoghurt	−0.03	0.831	−0.37	0.093	−0.07	0.596	−0.17	0.222
Dairy products **	−0.115	0.813	−0.173	**0.023**	−0.023	0.101	−0.125	**0.035**

Linear regression analysis adjusted for maternal education. Significant differences (*p* < 0.05) are shown in bold font. * Soft drinks and juices. ** Milk and yoghurt. BMI (body mass index) z-score was calculated according to the cut points of Cole et al. [24].

**Table 3 children-09-01947-t003:** Logistic regression analysis between the different components of the Mediterranean diet adherence questionnaire and the probability of having overweight/obesity (odds ratios and *p*-values) (*n* = 1075).

	3–6 y				6–12 y			
Ref: Yes	Boys (*n* = 139; 13%)		Girls (*n* = 141; 13.2%)		Boys (*n* = 390; 36.2%)		Girls (*n* = 405; 37.6%)	
	OR *	*p*	OR *	*p*	OR *	*p*	OR *	*p*
1. Have a fruit or a natural juice every day	2.807	0.082	0.837	0.832	1.436	0.267	1.048	0.893
2. Have a second piece of fruit every day	1381	0.536	1.274	0.676	1.188	0.541	0.815	0.471
3. Eat fresh or cooked vegetables once a day (1 time/day)	0.815	0.722	1.922	0.299	1.321	0.323	0.818	0.519
4. Eat fresh or cooked vegetables more than once a day (≥1 time/day)	1.525	0.435	1.587	0.427	1.145	0.627	1.228	0.466
5. Eat fish (2–3 times/week)	0.419	0.280	0.569	0.607	0.896	0.736	1.206	0.554
6. Visit a fast-food restaurant once or more a week (≥1 time/week)	0.478	0.174	1.719	0.528	1.223	0.494	0.845	0.591
7. Eat legumes more than one time a week (≥1 time/week)	0.325	0.298	1.875	0.404	0.634	0.257	1.648	0.163
8. Eat pasta or rice (5 days or more a week) (≥5 times/week)	1.431	0.500	1.750	0.375	1.028	0.920	0.745	0.288
9. Have a cereal or derivative for breakfast	1.867	0.217	1.213	0.743	1.612	0.102	0.876	0.679
10. Eat nuts (2–3 times/week)	1.373	0.558	0.756	0.649	1.336	0.308	2.901	**0.001**
11. Use olive oil at home	0.000	1.000	2.567	0.158	0.380	0.388	0.646	0.621
12. Have breakfast	1.360	0.798	0.000	0.999	0.590	0.460	3.006	0.108
13. Have dairy for breakfast	0.855	0.901	0.418	0.424	1.101	0.870	1.619	0.307
14. Have a breakfast of industrial pastries, cookies, cupcakes	0.688	0.471	2.140	0.270	1.164	0.572	1.161	0.596
15. Take two yoghurts and/or 40 g cheese every day	0.965	0.947	0.696	0.525	0.889	0.669	0.800	0.430
16. Take sweets and/or candies several times a day	0.424	0.153	0.913	0.916	1.042	0.937	0.803	0.605
	B **	*p*	B **	*p*	B **	*p*	B **	*p*
Total score MD	0.009	0.864	0.004	0.929	−0.013	0.813	−0.008	0.614

* Logistic regression analysis between the different components of the Mediterranean diet adherence questionnaire and the probability of having overweight/obesity. Significant differences (*p* < 0.05) are shown in bold font. ** Linear regression analysis adjusted for maternal education. MD, Mediterranean diet; Ref, reference group: Yes; OR, odds ratio. BMI (body mass index) z-score was calculated according to the cut points of Cole et al. [24].

**Table 4 children-09-01947-t004:** Association between total screen time categories and BMI z-score (*n* = 1075).

	3–6 y				6–12 y			
	Boys (*n* = 139; 13%)		Girls (*n* = 141; 13.2%)		Boys (*n* = 390; 36.2%)		Girls (*n* = 405; 37.6%)	
	B	*p*	B	*p*	B	*p*	B	*p*
Sleep duration weekday (hours)	−0.106	0.247	0.093	0.421	−0.147	**0.035**	−0.142	**0.040**
Sleep duration weekend (hours)	−0.221	**0.009**	0.101	0.286	−0.187	**0.001**	0.001	0.889
TST weekday (hours)	0.019	**0.020**	0.001	0.661	0.001	0.138	0.021	**0.007**
TST weekend (hours)	0.401	**0.035**	0.001	0.246	0.000	0.309	0.001	0.289
PA in club (min/day)	−0.100	0.740	0.001	0.920	0.001	0.129	0.001	0.653
Go and back walking to school (min/day)	−0.008	0.242	−0.007	0.549	−0.016	0.119	0.007	0.374

Linear regression analysis adjusted for maternal education. Significant differences (*p* < 0.05) are shown in bold font. TST, total screen time; PA, physical activity. BMI (body mass index) z-score was calculated according to the cut points of Cole et al. [24].

**Table 5 children-09-01947-t005:** Logistic regression analyses between different questions about the emotional well-being and self-esteem of children and BMI z-score (odds ratios and *p*-values) (*n* = 1075).

Ref: Always	3–6 y				6–12 y			
In the Last Week	Boys (*n* = 139; 13%)		Girls (*n* = 141; 13.2%)		Boys (*n* = 390; 36.2%)		Girls (*n* = 405; 37.6%)	
	OR	*p*	OR	*p*	OR	*p*	OR	*p*
My child was laughing and had a lot of fun	0.000	0.999	0.000	1.000	0.860	0.727	2.576	**0.043**
My child did not feel good about doing anything	3.429	0.999	0.390	0.473	0.396	0.206	0.589	0.579
My child felt lonely	2.048	0.368	2.057	0.407	0.402	0.373	0.585	0.574
My child felt insecure and anxious	4.026	0.999	2.567	0.158	1.007	0.993	0.859	0.838
My child was proud of himself/herself	0.401	0.397	4.714	0.056	0.734	0.346	1.335	0.388
My child felt on top of the world	1.074	0.892	0.703	0.594	0.751	0.364	0.993	0.981
My child felt good about himself	0.428	0.435	1.588	0.591	1.437	0.278	2.063	**0.040**
My child had good ideas	0.290	0.247	3.773	0.060	1.262	0.522	1.575	0.186
My child felt comfortable with us as parents	0.000	0.999	8.333	0.146	0.719	0.588	1.678	0.391
My child was comfortable at home	0.000	0.999	0.000	0.999	1.671	0.400	1.309	0.715
We argue at home	1.200	0.871	2.79	0.999	1.096	0.858	0.451	0.141
My child felt overprotected by us	0.760	0.643	2.057	0.317	0.502	**0.025**	0.916	0.783

Linear regression analysis adjusted for maternal education. Significant differences (*p* < 0.05) are shown in bold font. Ref, reference group: Always. BMI (body mass index) z-score was calculated according to the cut points of Cole et al. [24].

## Data Availability

The data presented in this study are available on request from the corresponding author.

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
