# Peer review of "Lifestyle Risk Factors for Overweight/Obesity in Spanish Children"

_children, 2022, doi:10.3390/children9121947_

Round 1
Reviewer 1 Report
The paper study on the influential factors of obesity for the children in Spanish. With the sample of 1075 observations, the research illustrates some interesting factors such as the food consumption, lifestyle, have either positive or negative impacts on the obesity. Generally, this topic is interesting and has policy implications, and the structure of this paper is also clear.
Some minor comments are as follows
The linear regressions are not good choice to reveal the relationships between the influential factors and binary outcome, due to the limited prediction power.
The robustness check of models is needed to be done and present in present paper. Moreover, there are many factors are highly multicollinearity that would lead to the insignificant results as the model become inefficient.
Proofreading is highly recommended for the publication consideration.
Author Response
“Lifestyle risk factors for overweight/obesity in Spanish children.”
Enclosed you will find a revision of our manuscript “Lifestyle risk factors for overweight/obesity in Spanish children.”. We would like to thank the reviewers for their thoughtful and constructive comments. We have considered all of the suggestions and have incorporated them into the revised manuscript (highlighted in yellow). We believe our manuscript is stronger as a result of these modifications. An itemized point-by-point response to the reviewers’ comments is presented below.
This manuscript contains material that is original and not previously published in text or on the Internet, nor is it being considered elsewhere until a decision is made as to its acceptability by the Children Review Board.
Reviewer #1 (Comments to the Author)
Open Review
( ) I would not like to sign my review report
(x) I would like to sign my review report
English language and style
( ) English very difficult to understand/incomprehensible
( ) Extensive editing of English language and style required
( ) Moderate English changes required
(x) English language and style are fine/minor spell check required
( ) I don't feel qualified to judge about the English language and style
|
|
Yes |
Can be improved |
Must be improved |
Not applicable |
|
Does the introduction provide sufficient background and include all relevant references? |
(x) |
( ) |
( ) |
( ) |
|
Are all the cited references relevant to the research? |
(x) |
( ) |
( ) |
( ) |
|
Is the research design appropriate? |
(x) |
( ) |
( ) |
( ) |
|
Are the methods adequately described? |
( ) |
(x) |
( ) |
( ) |
|
Are the results clearly presented? |
( ) |
(x) |
( ) |
( ) |
|
Are the conclusions supported by the results? |
(x) |
( ) |
( ) |
( ) |
Comments and Suggestions for Authors
The paper study on the influential factors of obesity for the children in Spanish. With the sample of 1075 observations, the research illustrates some interesting factors such as the food consumption, lifestyle, have either positive or negative impacts on the obesity. Generally, this topic is interesting and has policy implications, and the structure of this paper is also clear.
Some minor comments are as follows
The linear regressions are not good choice to reveal the relationships between the influential factors and binary outcome, due to the limited prediction power.
The robustness check of models is needed to be done and present in present paper. Moreover, there are many factors are highly multicollinearity that would lead to the insignificant results as the model become inefficient. Proofreading is highly recommended for the publication consideration.
Answer
Thank you for your comment. The linear and logistic regression models were selected considering the characteristics of the independent variable and they are able to identify the odds to have overweight/obesity according to the different variables of children's lifestyles that are the objective of this article.

Reviewer 2 Report
Thank you very much for allowing me to review this manuscript titled "Lifestyle risk factors for overweight/obesity in Spanish children." The authors have developed an exciting study with a large sample of children. However, some issues do not allow me to approve the manuscript in its current form.
Introduction
- Why is this study needed? Is this study contributing to some public food policy? Is there no information about this in Spain?
- Why did the authors select those behaviors (diet, physical activity, sleep duration, and others) selected? Diet seems well argued, but others no.
- what do the authors understand as lifestyle factors?
Methods
- From this section, it needs to be clarified who answered the questionnaires. In the limitations section, the authors declare that parents answered the questionnaires. I suggest adding who answered the questionnaire (father, mother, grandparents, among others) into the methods section.
- In line 64, it appears "Theschool participation rate…" It should say "The school…"
- Why were more 6-12 years old children participating than preschool children?
- Lines 84-85. Please add a reference to this protocol.
- In the methods, it appears more variables than what the authors mention in the introduction: screen time, self-esteem, and emotional well-being. Please, give some arguments about what these other variables should be incorporated in the introduction. However, it is necessary to consider that, for example, emotional well-being and self-esteem are not behaviors as the others.
- In the beverage consumption variables, what is the difference between the "(2) soft drinks (sugar-sweetened and light bever-94 ages)" and (4) "sugar-sweetened beverages (soft drinks 95 and juices)"?
Results
- I suggest adding some marks to the significative results in the tables to facilitate finding those results.
Discussion
- Review how this section was written. First, add the result you will discuss and then the comparison with the previous literature. For example, the discussion about the Mediterranean diet starts with a revision, and it is unclear what result is being discussed.
- It would be interesting to discuss why some of the results were not significant either. For example, what is the authors' opinion about not having significance with the Mediterranean diet variables?
- In lines 274 and 275, the authors state, "In addition, the low response rate on the socio-demographic and socioeconomic characteristics of the families as answering these questions can be complicated in vulnerable families." However, methods should be tested previously in the target population and adapted them to get the best information possible. Also, I do not understand why the authors indicate "the low response rate on the socio-demographic and socioeconomic." In Figure 1, it is indicated that 1152 questionaries for these variables were answered.
Author Response
“Lifestyle risk factors for overweight/obesity in Spanish children.”
Enclosed you will find a revision of our manuscript “Lifestyle risk factors for overweight/obesity in Spanish children.”. We would like to thank the reviewers for their thoughtful and constructive comments. We have considered all of the suggestions and have incorporated them into the revised manuscript (highlighted in yellow). We believe our manuscript is stronger as a result of these modifications. An itemized point-by-point response to the reviewers’ comments is presented below.
This manuscript contains material that is original and not previously published in text or on the Internet, nor is it being considered elsewhere until a decision is made as to its acceptability by the Children Review Board.
Reviewer #2 (Comments to the Author)
Open Review
Open Review
(x) I would not like to sign my review report
( ) I would like to sign my review report
English language and style
( ) English very difficult to understand/incomprehensible
( ) Extensive editing of English language and style required
( ) Moderate English changes required
( ) English language and style are fine/minor spell check required
(x) I don't feel qualified to judge about the English language and style
|
|
Yes |
Can be improved |
Must be improved |
Not applicable |
|
Does the introduction provide sufficient background and include all relevant references? |
( ) |
( ) |
(x) |
( ) |
|
Are all the cited references relevant to the research? |
(x) |
( ) |
( ) |
( ) |
|
Is the research design appropriate? |
(x) |
( ) |
( ) |
( ) |
|
Are the methods adequately described? |
( ) |
(x) |
( ) |
( ) |
|
Are the results clearly presented? |
( ) |
(x) |
( ) |
( ) |
|
Are the conclusions supported by the results? |
( ) |
(x) |
( ) |
( ) |
Comments and Suggestions for Authors
Thank you very much for allowing me to review this manuscript titled "Lifestyle risk factors for overweight/obesity in Spanish children." The authors have developed an exciting study with a large sample of children. However, some issues do not allow me to approve the manuscript in its current form.
Introduction
-Why is this study needed? Is this study contributing to some public food policy? Is there no information about this in Spain?
Answer: Thanks for your comments. The topics included in the manuscript were widely studied but there is no report on all the potential lifestyle determinants together and we aimed to analyze the sample prior to the intervention that has been implanted in the schools of Spain whose methodology is innovative and future papers will be published. Moreover, we consider different dimensions previously associated with childhood obesity, but also including the emotional well-being and self-esteem. These concepts are not usually evaluated in epidemiologic studies in relation with lifestyle and obesity. Moreover, few papers consider all these dimensions together in the same paper.
-Why did the authors select those behaviors (diet, physical activity, sleep duration, and others) selected? Diet seems well argued, but others no.
- What do the authors understand as lifestyle factors?
Answer: Thank you for your suggestion. We have considered these behaviors because they are considered as lifestyle behaviours and they have been studied extensively in epidemiology of obesity and other non-communicable diseases. We believe that lifestyles are associated with behaviors that people decide/adopt and that can be hazardous for their health.
Methods
-From this section, it needs to be clarified who answered the questionnaires. In the limitations section, the authors declare that parents answered the questionnaires. I suggest adding who answered the questionnaire (father, mother, grandparents, among others) into the methods section.
Answer: Thank you for your comment. We have added in the methods section the following sentence in Line 85
- In line 64, it appears "The school participation rate…" It should say "The school…"
Answer: Thanks. The sentence is ok now in line 78.
-Why were more 6-12 years old children participating than preschool children?
Answer: The participation was voluntary; the families have the option to refuse their involvement. The recruitment process was the same in all schools, therefore all families received the same information and they can decide their participation. One of the reasons for these differences in participation could be that the parents from school-age children´s could be more active in their relationship with the school activities. Also, the number of children who attend the school is lower than in the school age group.
-Lines 84-85. Please add a reference to this protocol.
Answer: Thanks. The reference has been added.
-In the methods, it appears more variables than what the authors mention in the introduction: screen time, self-esteem, and emotional well-being. Please, give some arguments about what these other variables should be incorporated in the introduction. However, it is necessary to consider that, for example, emotional well-being and self-esteem are not behaviors as the others.
Answer: The introduction has been completed with the relevance of self-esteem, and emotional well-being in children (lines 60 to 68).
“Weight-related impairment in the emotional well-being in young population has increase their significance in obesity research [23, 24]. The association between obesity and emotional well-being had showed inconsistent findings in children, adolescents, and adults [25-27]. However, the obesity has been found to be associated with low self-esteem, depressive mood in some studies [23, 28,29], in other studies no association has been observed [24,30,31]. Recently, the effects of weight-related body dissatisfaction in boys and girls during adolescence have been studied [32]. This situation is associated with negative body image which may be a key approach in obesity prevention and treatment programs”.
-In the beverage consumption variables, what is the difference between the "(2) soft drinks (sugar-sweetened and light beverages)" and (4) "sugar-sweetened beverages (soft drinks and juices)"?
Answer: We have considered to assess the different effect of soft drinks (carbonated beverages) and the juices, and also, to assess the effect of both groups (soft drinks and artificially juices) with a variable called sugar-sweetened beverages. The legend explain these issues.
Results
-I suggest adding some marks to the significative results in the tables to facilitate finding those results.
Answer:
Comments appreciated. The bold font has been applied to show the significative results in the tables.
Discussion
- Review how this section was written. First, add the result you will discuss and then the comparison with the previous literature. For example, the discussion about the Mediterranean diet starts with a revision, and it is unclear what result is being discussed.
Answer: Thank you for your comment. We have moved the paragraph where we describe our results before the paragraph where comparing with the literature.
-It would be interesting to discuss why some of the results were not significant either. For example, what is the authors' opinion about not having significance with the Mediterranean diet variables?
Answer: Thank you for your comment. The benefits in terms of obesity are attributable to a dietary pattern inspired by the principles of the Mediterranean Diet and not individual foods or components of the Mediterranean Diet. These have been associated with numerous health benefits and have been demonstrated to exert a preventive effect towards numerous pathologies, including obesity. In fact, studies that try to link a single food or food group with obesity do not provide significant results (1).
-In lines 274 and 275, the authors state, "In addition, the low response rate on the socio-demographic and socioeconomic characteristics of the families as answering these questions can be complicated in vulnerable families." However, methods should be tested previously in the target population and adapted them to get the best information possible. Also, I do not understand why the authors indicate "the low response rate on the socio-demographic and socioeconomic." In Figure 1, it is indicated that 1152 questionaries for these variables were answered.
Answer: Thank you for your suggestion. We agree with you and we believe that we should not mention the vulnerabilities as limitations so that we have not studied these variables in this article. For this reason, we have removed this sentence.
- Togo P, Osler M, Sørensen T, Heitmann B. Food intake patterns and body mass index in observational studies. International Journal of Obesity. 2001;25(12):1741-51.

Round 2
Reviewer 2 Report
Thank you for allowing me to review this new version of the manuscript titled "Lifestyle risk factors for overweight/obesity in Spanish 2 children." This version has been improved according to the reviewers' comments. However, I still cannot accept it in its current form because it is necessary to add why this study is needed in the introduction. Also, it is necessary to add to the introduction what the authors understand by lifestyle. I insist that the study has several variables that probably are consequences of the children and their families' lifestyles.
Author Response
Enclosed you will find a revision of our manuscript “Lifestyle risk factors for overweight/obesity in Spanish children.”. We would like to thank the reviewers for their thoughtful and constructive comments. We have considered all of the suggestions and have incorporated them into the revised manuscript (highlighted in yellow). We believe our manuscript is stronger as a result of these modifications. An itemized point-by-point response to the reviewers’ comments is presented below.
This manuscript contains material that is original and not previously published in text or on the Internet, nor is it being considered elsewhere until a decision is made as to its acceptability by the Children Review Board.
Reviewer #2 (Comments to the Author)
Open Review
(x) I would not like to sign my review report
( ) I would like to sign my review report
English language and style
( ) English very difficult to understand/incomprehensible
( ) Extensive editing of English language and style required
( ) Moderate English changes required
( ) English language and style are fine/minor spell check required
(x) I don't feel qualified to judge about the English language and style
|
Yes |
Can be improved |
Must be improved |
Not applicable |
|
|
Does the introduction provide sufficient background and include all relevant references? |
( ) |
(x) |
( ) |
( ) |
|
Are all the cited references relevant to the research? |
(x) |
( ) |
( ) |
( ) |
|
Is the research design appropriate? |
(x) |
( ) |
( ) |
( ) |
|
Are the methods adequately described? |
(x) |
( ) |
( ) |
( ) |
|
Are the results clearly presented? |
(x) |
( ) |
( ) |
( ) |
|
Are the conclusions supported by the results? |
(x) |
( ) |
( ) |
( ) |
Comments and Suggestions for Authors
Thank you for allowing me to review this new version of the manuscript titled "Lifestyle risk factors for overweight/obesity in Spanish 2 children." This version has been improved according to the reviewers' comments. However, I still cannot accept it in its current form because it is necessary to add why this study is needed in the introduction.
Answer: Thank you for your suggestion. The topic included in the manuscript was widely studied but there is no report on all the potential lifestyle determinants together, including the emotional well-being and self-esteem in a Spanish sample whose methodology is innovative.
This paragraph has been added in the introduction section (lines 72-74).
Also, it is necessary to add to the introduction what the authors understand by lifestyle. I insist that the study has several variables that probably are consequences of the children and their families' lifestyles.
Answer: Thank you for your suggestion. The lifestyle is a summary of our behavior patterns that we have developed during our socialization processes which occur during the pre-school´s and school-age. These behaviours are learned in relationships with parents, peers, friends and siblings, or through the influence of school or the media. These patterns include children´s physical activity, screen and sleep time, beverages, and food consumption. On the other hand, our lifestyle is also influenced by our emotional well-being and self-esteem as main psychological and emotional factors to ensure a balance that support the daily decisions.
This paragraph has been added in the introduction section (lines 65-71).
